# An Update on the Efficacy of Single and Serial Intravenous Ketamine Infusions and Esketamine for Bipolar Depression: A Systematic Review and Meta-Analysis

**DOI:** 10.3390/brainsci13121672

**Published:** 2023-12-02

**Authors:** Nicolas A. Nunez, Boney Joseph, Rakesh Kumar, Ioanna Douka, Alessandro Miola, Larry J. Prokop, Brian J. Mickey, Balwinder Singh

**Affiliations:** 1Department of Psychiatry & Psychology, Mayo Clinic, 200 First Street SW, Rochester, MN 55905, USA; nunez.nicolas@mayo.edu (N.A.N.);; 2Department of Psychiatry, University of Utah, Salt Lake City, UT 84112, USA; 3Department of Neurology, Mayo Clinic, Rochester, MN 55905, USA; 4Department of Neuroscience (DNS), University of Padova, 35122 Padua, Italy; 5Mayo Medical Libraries, Mayo Clinic College of Medicine, Rochester, MN 55905, USA

**Keywords:** ketamine, treatment-resistant bipolar depression, metanalysis, systematic review

## Abstract

Ketamine has shown rapid antidepressant and anti-suicidal effects in treatment-resistant depression (TRD) with single and serial intravenous (IV) infusions, but the effectiveness for depressive episodes of bipolar disorder is less clear. We conducted an updated systematic review and meta-analysis to appraise the current evidence on the efficacy and tolerability of ketamine/esketamine in bipolar depression. A search was conducted to identify randomized controlled trials (RCTs) and non-randomized studies examining single or multiple infusions of ketamine or esketamine treatments. A total of 2657 articles were screened; 11 studies were included in the systematic review of which 7 studies were included in the meta-analysis (five non-randomized, N = 159; two RCTs, N = 33) with a mean age of 42.58 ± 13.1 years and 54.5% females. Pooled analysis from two RCTs showed a significant improvement in depression symptoms measured with MADRS after receiving a single infusion of ketamine (1-day WMD = −11.07; and 2 days WMD = −12.03). Non-randomized studies showed significant response (53%, *p* < 0.001) and remission rates (38%, *p* < 0.001) at the study endpoint. The response (54% vs. 55%) and remission (30% vs. 40%) rates for single versus serial ketamine infusion studies were similar. The affective switch rate in the included studies approximated 2.4%. Esketamine data for bipolar depression are limited, based on non-randomized, small sample-sized studies. Further studies with larger sample sizes are required to strengthen the evidence.

## 1. Introduction

Bipolar disorder (BD) is a recurring mental health condition characterized by alternating mood fluctuations, ranging from periods of high-energy mania or hypomania to severe depressive episodes and subsyndromal depression. The subsyndromal/subthreshold symptoms can lead to substantial impairment in daily functioning and contribute to significant morbidity and mortality [1]. In the United States, the estimated lifetime prevalence rates for different types of bipolar disorder are as follows: 1.0% for bipolar type I disorder (BD-I), 1.1% for bipolar type II disorder (BD-II), and 2.4% for subthreshold bipolar disorder [2]. Subsyndromal depressive symptoms cause significant functional impairment and are also a risk factor for the recurrence of depressive episodes.

Despite the FDA’s approval of new atypical antipsychotic medications for bipolar depression, including cariprazine and lumateperone, there are still ongoing challenges related to their effectiveness and overall treatment outcomes [3]. Current pharmacological options for bipolar depression include cariprazine, lumateperone, lurasidone, quetiapine, and olanzapine/fluoxetine combination. Lamotrigine, as a maintenance therapy, has shown the ability to reduce the risk of future depressive episodes, although it takes almost 6–8 weeks to reach therapeutic dosage. Augmentation strategies with mood stabilizers, thyroid hormones, and dopamine agonists such as pramipexole and psychostimulants have attempted to surpass the significant challenges of the depressive phase of the illness. The delayed onset of therapeutic effects, coupled with limited response and remission rates, presents a significant challenge in the treatment of bipolar depression. Notably, over one-third of patients do not respond to treatment interventions, which is commonly known as treatment-resistant bipolar depression (TRBD) [4]. The absence of faster acting antidepressants creates a significant challenge which significantly impacts the quality of life and overall functioning of individuals with TRBD. The exploration and development of innovative compounds with rapid-acting antidepressant properties is a priority both for clinicians and researchers [5].

One such compound has been ketamine, an anesthetic agent modulating the gamma-aminobutyric acid (GABA)ergic and glutamatergic system, mechanistically acting as an N-methyl D-aspartate (NMDA) receptor antagonist, which has been now repurposed as a rapid-acting antidepressant for individuals with treatment-resistant depression (TRD) [6,7] with some studies showing a reduction in suicidal ideation as fast as within one day [8]. A recent systematic review showed improvement in depressive symptoms after receiving a single infusion of ketamine, including improvement in anhedonia and suicidal ideation items, although these improvements were short lived for approximately 3 days [9]. While the preliminary findings suggest that ketamine may have a role in the treatment of TRBD, it is important to emphasize that the current evidence base supporting ketamine’s use for bipolar depression is limited. This underscores the urgent need for additional data and research to establish the full potential and long-term efficacy of ketamine for individuals with TRBD.

Additionally, the S-enantiomer-esketamine is now FDA approved in adults for unipolar TRD (as an augmentation agent with an oral antidepressant) and for major depressive disorder (MDD) with suicidal ideations or suicidal behaviors [10,11]. Although esketamine is not FDA approved for suicidal ideation, different studies support the use of intranasal esketamine for MDD with suicidal thoughts or actions (MDSI). In esketamine drug trials, patients with bipolar disorders were excluded; thus, there are a lack of data from randomized controlled trials (RCTs) regarding esketamine’s efficacy and safety for TRBD. Racemic ketamine and esketamine are being extensively prescribed and offered as off-label treatments for TRBD; mechanistically, its impact on molecular pathways needs to be elucidated. In a previous systematic review and meta-analysis encompassing 56 studies with a total of 2801 participants, examining individual biomarkers and their relationship with responses to ketamine or esketamine treatment, no association was found between treatment response and baseline blood biomarkers [12]. However, when a longitudinal analysis was conducted, it was revealed that only those individuals who responded favorably to ketamine exhibited an increase in brain-derived neurotrophic factor compared to their pre-treatment levels, while non-responders did not demonstrate this change.

Particularly of interest is the comparison of the literature examining single compared to serial infusions of intravenous (IV) ketamine administration and its overall efficacy and tolerability. Previous studies have suggested that serial ketamine infusions seem to be more effective than single infusions for depression and had similar side effects rates [13]. In a study of 24 TRD patients of which 21 received six ketamine infusions (dose 0.5 mg/kg), 71% achieved response; interestingly, early response after 4 h of infusion predicted response status at the end of serial infusions [14]. However, in the real-world setting, the response and remission rates with serial ketamine and esketamine treatments are much lower [15]. There are a lack of data regarding dose escalation and the impact of ketamine’s efficacy, although some open-label studies have implemented such strategies with mixed results [16,17].

Considering a lack of RCTs underscoring the effectiveness of ketamine and esketamine for TRBD, we conducted an updated systematic review and meta-analysis to appraise the current evidence from the literature on the efficacy and tolerability of single and multiple infusions of ketamine and esketamine treatments in bipolar depression.

## 2. Methods

This meta-analysis and systematic review was performed in accordance with the Preferred Reporting Items for Systematic reviews and Meta-Analyses (PRISMA) guidelines [18]. The study protocol was registered with the open science framework (https://osf.io/ext4d (accessed on 10 November 2023)).

**Figure 1 brainsci-13-01672-f001:**
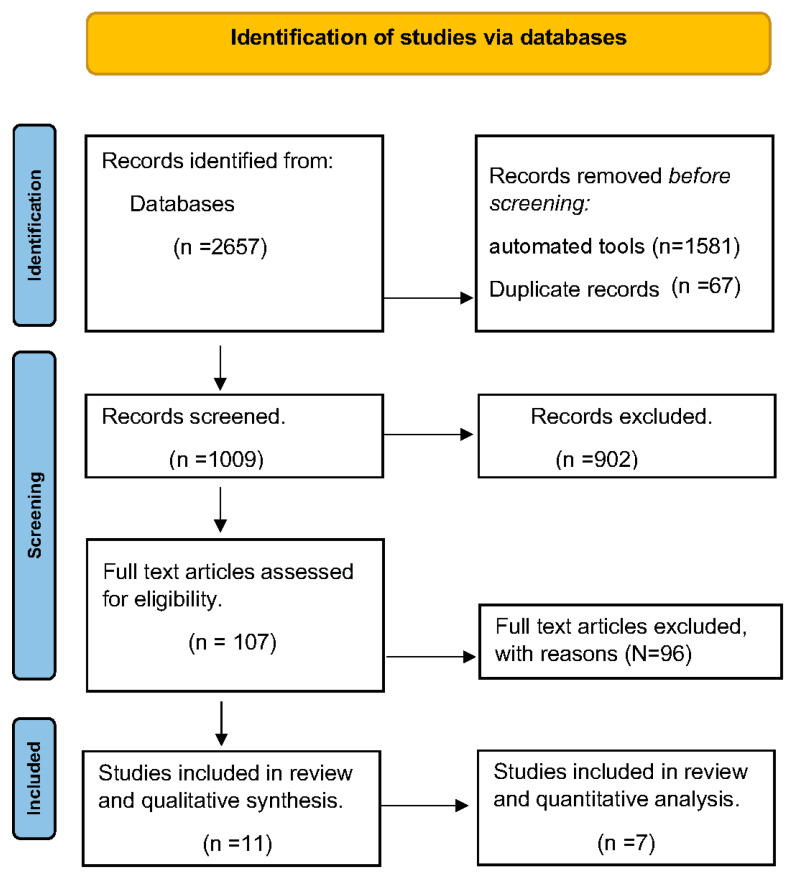
Study flow diagram.

### 2.1. Data Sources and Search Strategies

A thorough exploration of various databases was conducted, covering the period from the inception of each database to 6 February 2023, without imposing any language or date constraints. The databases included in the search were MEDLINE(R) and Epub Ahead of Print, In-Process and Other Non-Indexed Citations, and Daily, Ovid EMBASE, Ovid PsycINFO, Ovid Cochrane Central Register of Controlled Trials, and Scopus. The search specifically targeted RCTs or non-randomized studies/open-label trials focusing on the single or multiple infusion of ketamine/esketamine in adult subjects with bipolar depression or TRBD. The search strategy, formulated by an experienced medical librarian (LJP) in collaboration with the study investigators, employed a combination of controlled vocabulary and keywords. Additionally, the references of potentially eligible articles were scrutinized to broaden the search. The detailed strategy, listing all search terms and their combinations, is accessible in Appendix A of the Supplementary Material.

### 2.2. Study Selection

Two reviewers, namely NAN and BJ, worked both independently and in pairs to identify and screen the titles and abstracts of studies that met our inclusion criteria. The inclusion criteria were as follows: (1) *Population*—adult (≥18 years) patients diagnosed with bipolar depression/TRBD; (2) *Intervention*—pharmacological treatment with ketamine (single or serial infusions) or esketamine; (3) *Control*—placebo or control or treatment as usual; (4) *Outcome*—reported outcome data on remission and response rates, all-cause of discontinuation and change in severity of depressive symptoms assessed by the use of standardized behavioral scales (such as Brief Psychiatric Rating Scale (BPRS), Montgomery–Åsberg Depression Rating Scale (MADRS); Hamilton Depression Rating Scale (HAMD-17)) from baseline to endpoint; and reported side effects. Studies were considered irrespective of the language. We included RCTs, non-randomized studies, and observational studies. Animal studies, case series, case reports, case-control studies, cohort studies not focused on patients with bipolar depression, and commentaries, conceptual papers, editorials and book chapters were excluded from the systematic review. The included articles underwent qualitative analysis to assess efficacy, tolerability, dose range, duration of treatment and study biases.

### 2.3. Data Collection

Data from the included studies was extracted by four reviewers (NAN, BJ, RK, and ID) using a standardized data extraction form. We extracted the following information from the included studies: study characteristics (first author’s name, year, country), study design, sample size, total patients (females %), age (years), outcome measures and conclusions. The collected data were entered in an Excel file, and the disagreements were cleared through discussions.

### 2.4. Methodological Quality and Risk of Bias Assessment

The assessment of risk of bias for the RCTs involve the utilization of the Cochrane Collaboration’s risk of bias tool, which evaluated six domains: allocation concealment, sequence generation, blinding of study participants and personnel, blinding of the outcome assessment, incomplete outcome data, selective reporting, and other biases [19]. A high risk of bias was assigned if the described protocols raised concerns about bias in a specific domain. If descriptions of the domain were omitted from the primary text, risk was labeled as “unknown”. Conversely, if an adequate protocol was described for a given domain, it would be designated as “low risk”. For the open-label/non-randomized studies, the methodological quality was assessed using *Methodological Index for Non-Randomized Studies* (MINORS) [20]. In Appendix A, we provided the risk of bias assessment and MINORS scores, respectively. Publication bias was not assessed due to the limited number of studies (<10 studies) [21].

### 2.5. Statistical Analyses

Continuous variables were reported as mean ± standard deviations (SD), and categorical variables were reported as frequency and proportions. Data for response or remission were analyzed. For the RCT studies, we computed the weighted mean difference (WMD) for each variable analyzed and combined the effect sizes across studies to provide overall estimates and their 95% confidence intervals (CIs). All the statistical analyses were conducted using the “meta” and “metaprop” software packages of R software (version 4.2.2) in RStudio(version 1.1.463, Mozilla/5.0 (Macintosh; Intel Mac OS X 10_16) AppleWebKit/605.1.15 (KHTML, like Gecko) USA) using the DerSimonian(R studio version 1.1.463) and Laird random effects model to summarize effect sizes and pooled prevalence with logit transformation [22,23]. Between-study heterogeneity was assessed by the Cochran’s Q test and quantified with the I^2^ statistic [24,25]. According to the Cochrane handbook, an I^2^ between 30% and 60% indicates moderate heterogeneity and between 50% and 90% substantial heterogeneity [19].

## 3. Results

### 3.1. Study Selection

A total of 2657 articles were screened of which 107 were selected for full text review (Figure 1). Eleven studies were included in this systematic review of which nine studies reported change in depression scores post-ketamine infusions [16,26,27,28,29,30,31,32,33] and two studies reported data on esketamine [34,35]. Only two RCTs [26,27] comparing single-infusion IV racemic ketamine to placebo/control were included for the primary analysis to avoid a high risk of bias [32]. Kappa for interobserver agreement during both phases of study selection was high (>95%) [36].

### 3.2. Characteristics of Included Studies

A total of eleven studies were included in the systematic review and seven studies were included in the meta-analysis (five non-randomized studies, N = 159; two RCTs, N = 33), mean age 42.58 ± 13.1 years and 54.5% were females. With regard to route of administration, two observational studies examined the use of intranasal esketamine [34] and subcutaneous esketamine [35] in patients with TRBD; the nine remaining studies examined administered IV ketamine concomitant with an existing mood stabilizer/antidepressant (Table 1).

For the two RCTs included in the meta-analysis, pooled analysis showed significant improvement in depression symptoms measured by the MADRS (WMD at 1-day = −11.07; 95% CI = −12.3, −9.9, and WMD at 2 days = −12.03; 95% CI = −13.24, −10.82) after receiving a single infusion of ketamine. The open-label studies showed significant response (four studies, 53%, CI = 29–75%, *p* <0.001) and remission rates (five studies, 38%, CI = 18–63%, *p* < 0.001) at the study endpoint (Figure 2A,B). In Figure 3A,B, we illustrate the response and remission rates for RCTs and open-label studies.

Overall, single infusions response rates (three studies, 54%, 95% CI = 31–75%, *p* < 0.001) and remission rates were significant (four studies, 30%, 95% CI = 17–47%, *p* < 0.001) at the study endpoint. Serial ketamine infusions (three non-randomized studies) showed a significant response rate (55%, CI = 15–89%, *p* < 0.001) and remission rates (40%, CI = 7–86%, *p* < 0.001) at 3–4 weeks (Figure 4 A,B). However, the response and remission rates did not significantly differ among the single and serial IV ketamine infusion studies (54% vs. 55% response rates, *p* = 0.913; 30% vs. 40% remission rates, *p* = 0.429, respectively). Additionally, included studies also reported improvement in suicidal ideation and anhedonia after IV ketamine infusions [27,32]. Zhou et al [30] did not provide response and remission rates in their study and was the only study where depressive symptoms worsened (at 3 weeks) after an initial improvement (at 7 days) with ketamine treatment. 

Of the two esketamine studies, Martinotti and colleagues in a 3-month observational study included 70 patients with either TRBD (n = 35) or TRD (n = 35) who received two doses of intranasal esketamine (range 28–84 mg) weekly for the first month and then one dosage weekly for the rest of the study duration. The authors reported no significant differences in esketamine effectiveness in TRD versus TRBD patients with a pronounced reduction in depressive symptoms after the first month (MADRS mean score reduction in the first month of the TRBD group: −13.03, TRD: −12.21). Additionally, it showed anxiolytic action in the TRBD group. They reported one case of affective switching in the TRBD group [34]. Delfino and colleagues examined the anti-anhedonic effects of subcutaneous esketamine in 70 patients with MDD (n = 39) or bipolar depression (n = 31) for 6 weeks. They reported a significant reduction in item 8 of the MADRS in both groups (*p* < 0.001) with no significant differences between groups and a higher reduction in scores predicted post-treatment anhedonia severity [35].

### 3.3. Methodological Quality and Risk of Bias Assessment

In Appendix A, we report the quality assessment of the included studies in the meta-analysis. All three RCTs were of good quality, although there is a concern for adequate blinding in the ketamine trials [26,27,32]. The open-label/observational studies did not include consecutive patients and did not provide a prospective calculation of power or study sample size, although they did not report loss to follow up <0.5% on patients. One study did not report attrition bias and did not adjust for confounders [28,29].

### 3.4. Adverse Events

Most of the included studies, single and serial infusions, reported non-severe adverse side effects such as drowsiness, dizziness, headache, or insomnia. Dissociative symptoms were reported during the administration of ketamine but not further after 40 min of infusion [16,26,27,32,33]. One participant in the RCTs experienced manic symptoms [26], and one developed hypomanic symptoms in an open label-observational study [29]. Three patients (4.5%) reported treatment-emergent affective switching (hypomania) in an observational study where 66 patients received four IV ketamine treatments for TRBD in escalating doses [16]. Overall, the incidence of manic/hypomanic symptoms in this review was 5 out of 208 patients for IV ketamine (2.4%, 95% CI 0.29–4.5%). A detailed description of the inclusion–exclusion criteria and adverse events of the included studies in this review is provided in Appendix A in the Supplementary Material.

## 4. Discussion

This comprehensive systematic review of eleven studies among patients with bipolar depression underscored that both single and serial infusions of IV racemic ketamine are similarly effective and with good tolerability. Overall, IV ketamine improved depressive scores significantly in both RCTs and non-randomized studies. Interestingly, among the serial infusions studies, three studies reported an increase in depressive symptoms within 1–2 weeks following the last treatment [30,31,33]. This potentially highlights the short duration of ketamine and raise concerns regarding the durability and efficacy in long-term treatments. However, there is still a need to define adequate dosages for maintenance and long-term treatment and a better phenotypic characterization of patients who will respond to maintenance treatment.

Of the included studies, two examined the intranasal [34] and subcutaneous [35] use of esketamine. Those studies suggest similar efficacy in TRD and TRBD as well as its anti-anhedonia effects. Although serial infusions (open label/observational studies) showed slightly higher response and remission rates when compared to single infusions, these differences were not significant; however, single infusions (mostly RCTs studies with lower effect sizes) may have had more stringent inclusion criteria or lower expectancy effects, which may have contributed to a slighter lower response rate. As an example, Fancy et al. allowed for a dosage increase after the first two infusions [16]. We should also note that there was a high heterogeneity in both groups of patients as well as utilizing different ketamine dosages, although most studies use subanesthetic dosages of 0.5 mg/kg over 40 min. This reported dose has been shown to be more effective in reducing depressive symptoms according to a meta-analysis of six trials when compared to lower dosages [37]. Furthermore, Fancy et al. provided information regarding BD subtypes, showing a trend toward a greater reduction in depressive symptoms in BD-II compared to BD-I [16].

Two of the RCTs included have shown ketamine’s anti-anhedonic effect and rapid improvement in suicidal ideation [27,32]. A recent study by Wilkowska and colleagues showed a significant reduction in anhedonia across ketamine infusions (*p* < 0.001) in patients with unipolar and bipolar treatment resistant depression; however, they did not provide separate data for patients with TRBD [38]. Reduction in anhedonia with ketamine is associated with reduction in suicidal thoughts independent of depressive symptoms. A recent study reported that an improvement in the SHAPS 24 h post-ketamine may account for a 13% variance in reduction in SI after ketamine administration [39]. Ketamine’s antianhedonic effect is an active area of investigation considering that preclinical studies suggest a synergistic effect of ketamine and lithium, highlighting activation of the mTOR pathway and GSK-3 inhibition, thus improving both depression–anhedonia and suicidality [40].

In the systematic review, the reported side effects were moderate for single and serial infusions. Importantly, considering the previous concern regarding the possibility of induction of manic/hypomanic symptoms in this population [41], the overall incidence in the included studies in this review approximated 2.4%. Nevertheless, we should be cautious about these conclusions considering the small sample sizes for most of the infusions.

### 4.1. Strengths and Limitations

Several limitations of these meta-analytic findings should be acknowledged. A major limitation is the small number of included studies, which overall underscores the pressing need for a greater development of RCTs. Additionally, all the serial infusion studies were open-label or non-randomized, lacking a control group, which presents another limitation. Secondly, the studies regarding efficacy and tolerability for TRBD did not differentiate in many cases between BD-I and BD-II; thus, we were unable to perform a subgroup analysis to ascertain whether ketamine’s efficacy differs between patients with BD-I compared to BD-II [42]. Third, except for two studies, most of the studies had a higher percentage of female participants, thus limiting the generalizability. However, data so far from prior RCTs suggest similar efficacy for ketamine among the males and females [43]. Fourth, we did not consider potential pharmacological interactions or analyzed the effect of baseline medications (i.e., mood stabilizers) in augmenting ketamine treatment. Fifth, due to the limited number of studies, we were unable to assess for publication bias.

### 4.2. Clinical Considerations

Patients with TRBD should carefully consider ketamine and esketamine for short duration while concurrently continuing a robust mood stabilizer in an acute depressive episode. The study by Fancy et al. showed a more robust response among patients with BD-II compared to BD-I, thus suggesting a possible option for BD-II. It is important to note that the prolonged use of ketamine carries a potential risk for treatment-emergent affective switching and risk of dependency, necessitating careful and thorough monitoring. Dose escalation studies beyond 0.5 mg/kg body weight for racemic ketamine have not been established for treating TRBD; thus, extreme caution is advisable if dose escalation is considered to minimize the risk of negative consequences. In the absence of an RCT investigating intranasal esketamine for TRBD, there are limited data to provide recommendation for esketamine use for TRBD, although the data from one observational study seem reassuring.

## 5. Conclusions

Findings of this study suggest the efficacy of single and serial IV ketamine infusions as a promising treatment for bipolar depression at least in the short term. There is a lack of serial infusion RCTs comparing ketamine to control for TRBD. Moreover, future studies comparing different ketamine/esketamine routes of administration, as well as different dosages and lengths of treatment, are needed to solidify the evidence. At the moment, the majority of serial infusion data are from non-randomized studies that are prone to bias. Adequately powered RCTs investigating the long-term efficacy and safety during the maintenance phase of IV ketamine are needed to strengthen and optimize the evidence base of ketamine for TRBD. Esketamine is not FDA approved for TRBD; thus, future RCTs should include patients with TRBD to identify the appropriate dosing schedule, efficacy and safety of esketamine for TRBD. Treatment-emergent affective switching is a concern with antidepressant use in bipolar disorder; thus, longitudinal studies would help identify the rates of switching with ketamine and esketamine in bipolar disorders.

## Figures and Tables

**Figure 2 brainsci-13-01672-f002:**
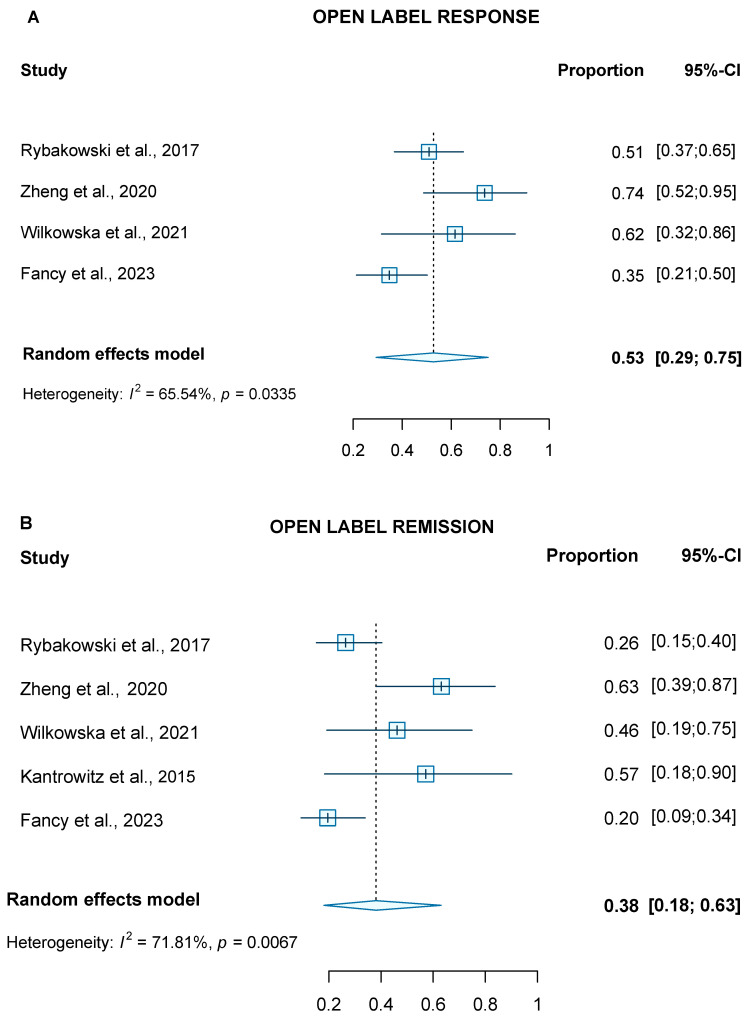
(**A**) Forest plot of pooled response rates for open label studies; (**B**) forest plot of pooled remission rates for open-label studies [16,28,29,31,33].

**Figure 3 brainsci-13-01672-f003:**
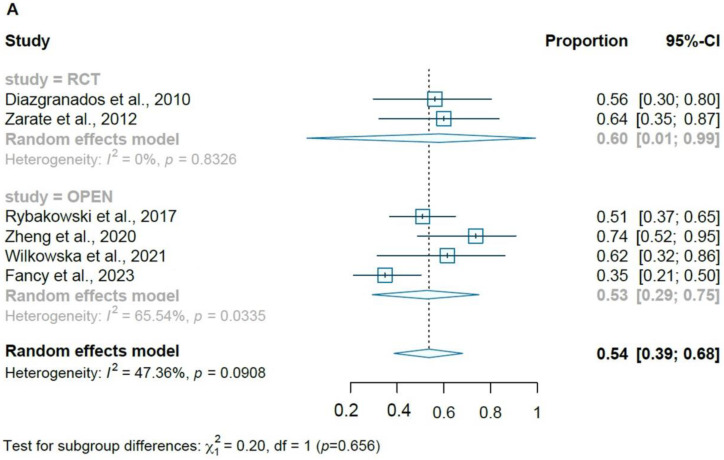
(**A**) Forest plot of pooled response rates for randomized controlled trials and open-label studies; (**B**) forest plot of pooled remission rates for randomized controlled trials and open-label studies [16,26,27,28,29,31,33].

**Figure 4 brainsci-13-01672-f004:**
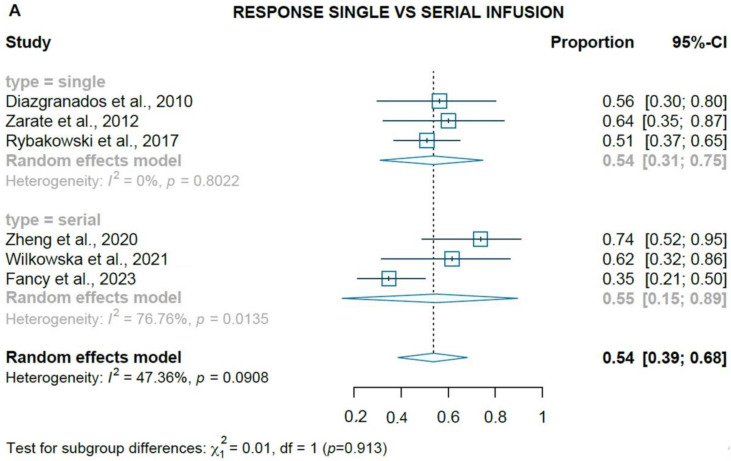
(**A**) Forest plot of pooled response rates for single vs. serial infusions; (**B**) forest plot of pooled remission rates for single vs. serial infusions [16,26,27,28,29,31,33].

**Table 1 brainsci-13-01672-t001:** Characteristics of the included studies.

References, Year	Type of Study	Total Patients, Females (n(%))	Age (Years) (Mean ± SD)	Number of Infusions	Outcome Measures	Conclusions
Diazgranados et al., 2010 [26]	RCT, Crossover (ketamine–saline placebo)	18, 12 (66.7)	47.9 (13.1)	one	HAMD-17, MADRS, BDI, YMRS, CADSS, BPRS	Single intravenous dose of ketamine resulted in a robust and rapid antidepressant effect
Zarate et al., 2012 [27]	RCT, Crossover (ketamine–saline placebo)	15, 8 (53.3)	46.7 (10.4)	one	HAMD-17, MADRS, BDI, YMRS, CADSS, BPRS	Rapid and robust antidepressant response and rapid improvement in suicidal ideation following a single intravenous dose of ketamine
Kantrowitz et al., 2015 [29]	Open label	8, 5 (62.5)	37 (16)	one	HAMD-17, BDI	25% or more improvement in HAM-D in 7 patients after ketamine infusion
Grunebaum et al., 2016 [32]	RCT (ketamine–midazolam)	16, 10 (62.5)	41.25 (12.45)	one	HAMD-17, SSI, BDI, POMS, YMRS, CGI	Suicidal thoughts were lower after ketamine than after midazolam at a trend level of significance. Reduction in depression scores not statistically significant in ketamine group when compared to midazolam
Rybakowski et al., 2017 [28]	Open label	53, 40 (75.5)	47 (12.6)	one	HAMD-17	Rapid antidepressant effect after single intravenous dose of ketamine infusion and a reduction of ≥50% on HAM-D in 24.5% patients at 24 h and in 51% patients on the seventh day after infusion
Zhou et al., 2020 [30]	Open label	38, 16 (42.1)	43.1 (5.3)	nine	HAMD-17, YMRS	Significant differences were observed in HAMD scores after one week of ketamine plus antidepressant treatment with an average reduction of 49.8% Patients at end of trial exhibited an increase in symptom severity relative to the baseline potentially related to a neural desensitization
Zheng et al., 2020 [31]	Open label	19, 6 (31.6)	35.8 (12.7)	six	MADRS, HAMD-17, BPRS, CDRS	Time to response and remission was 9 and 12 days, respectively, reaching rates of 73.7% and 63.2%. Improvement of depressive symptoms and suicidal ideation were sustained with subsequent infusions. No significant cardiovascular or psychomimetic side effects.
Wilkowska et al., 2021 [33]	Open label	13, 6 (76.9)	49.5 (15.1)	eight	MADRS, BPRS, CADSS, YMRS	IV ketamine was effective and well tolerated in the TRBD sample with response rates of 61.5% and remission of 46.2% following an average of 22.1 days to either response or remission. Side effects were mild (cardiovascular and psychomimetic) with no serious adverse events or affective switches
Delfino et al., 2021 [35]	Open label observational	70, 45 (64)	39.5 (12.3)	Six	MADRS (anhedonia)	Significant reduction in anhedonia after the first subcutaneous infusion and increased with repetitive infusions (*p* < 0.001) in both groups. No significant differences in esketamine anti-anhedonic effect. Subcutaneous administration seems to be equally effective in anti-anhedonic effects with good tolerability as simpler and inexpensive procedure.
Fancy et al., 2023 [16]	Open label observational	66, 39 (59.1)	45.7 (13.4)	four	QIDS-SR 16	There was an improvement of depressive and anxiety symptoms, as well as psychosocial functioning (family and social domains) after repeated IV infusions of ketamine with a 6.08 ± 1.39 reduction in QIDS-SR16. After four infusions response rates were of 35% and 20% for remission. Ketamine was well tolerated and treatment emergent hypomania was observed in 4.5% of the sample.
Martinotti et al., 2023 [34]	Open label observational	70, 38 (54.2)	52.7 (10.9)	2 weekly doses in the 1st month and 1 dose per week in the following 2 months.	MADRS, HAMD-17	Reduction in depressive symptoms starting after 1 month of intranasal esketamine with no significant differences between groups. No significant side effects in terms of affective switch (low risk for mania/hypomania). Patients with TRBD showed lower side effects than TRD (57.14% vs. 77.15%).

Abbreviations: MADRS: Montgomery–Åsberg Depression Rating Scale; HAMD-17: Hamilton Depression Rating Scale; BDI: Beck Depression Inventory; YMRS: Young Mania Rating Scale; CADSS: Clinician-Administered Dissociative States Scale; BPRS: Brief Psychiatric Rating Scale; SSI: Scale for Suicidal Ideation; POMS: Profile of Mood States; QIDS-SR 16: Quick Inventory for Depression Symptomatology-Self Report-16 (QIDS-SR16); CGI: Clinical Global Impression; RCT: Randomized Controlled Trial.

## Data Availability

Data are contained within the article or Appendix A.

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
