# Peer review of "An Update on the Efficacy of Single and Serial Intravenous Ketamine Infusions and Esketamine for Bipolar Depression: A Systematic Review and Meta-Analysis"

_brainsci, 2023, doi:10.3390/brainsci13121672_

Round 1
Reviewer 1 Report
Comments and Suggestions for Authors
This is an interesting manuscript on the ketamine use in bipolar depression. While the topic is of prime interest to the clinicians several ambiguities and brevities arise with regard to ketamine use in bipolar depression alike to major depressive disorder.
Minor remarks:
Introduction:
Please, briefly address lack of EMA approved treatments for acute bipolar depression. It might be useful to emphasize the esketamine-nasal spray (ESK-NS) as the approved formulation in line with the precise statement on the backbone antidepressants approved (SSRI/SNRI). Also, MDSI is FDA label for ESK-NS with MDD-PE on EMA party. It might be of an interest to the readers.
Methodology:
Please, indicate the HAM-D variant used in psychometrics (i.e. HAM-D-17). Moreover, please, clarify the study selection for ketamine/esketamine administration. Overall, it appears all apply to parenteral formulation for ketamine (iv, sc) and nasal spray for esketamine. It is, however, in between the lines.
Discussion:
Anhedonic effect was also replicated (doi: 10.3389/fpsyt.2021.704330) by Wilkowska et al. (2021) and might ameliorate the observation mentioned in the discussion.
Conclusions:
I strongly encourage elaborating on the implication for the future research with regard to dose, formulation, administration frequency and treatment duration.
Author Response
Dear reviewer,
We appreciate your efforts and the thoughtful critiques. We have addressed every comment to the best of our ability. Our point-by-point responses are listed below.
Reviewer 1:
Comments to the Author
This is an interesting manuscript on the ketamine use in bipolar depression. While the topic is of prime interest to the clinicians several ambiguities and brevities arise with regard to ketamine use in bipolar depression alike to major depressive disorder.
Minor remarks:
Introduction:
Please, briefly address lack of EMA approved treatments for acute bipolar depression. It might be useful to emphasize the esketamine-nasal spray (ESK-NS) as the approved formulation in line with the precise statement on the backbone antidepressants approved (SSRI/SNRI). Also, MDSI is FDA label for ESK-NS with MDD-PE on EMA party. It might be of an interest to the readers.
Response: Thank you for this comment. We have added to the introduction the following text starting at -page 2, line 83-: “Although esketamine is not FDA approved for suicidal ideation, different studies support the use of intranasal esketamine for MDD with suicidal thoughts or actions (MDSI). In esketamine trials, patients with bipolar disorders were excluded, thus, lack of data from randomized controlled trials regarding esketamine’s efficacy and safety for TRBD”.
Methodology:
Please, indicate the HAM-D variant used in psychometrics (i.e. HAM-D-17). Moreover, please, clarify the study selection for ketamine/esketamine administration. Overall, it appears all apply to parenteral formulation for ketamine (iv, sc) and nasal spray for esketamine. It is, however, in between the lines.
Response: Thank you for this observation. We have now added the correct variant of the behavior scale HAMD-17-page 4, line 152-as well as in table 1. We have also clarified the formulation for ketamine/esketamine in the results section-page 5 line 287- which now reads: “ With regards to route of administration, two observational studies examined the use of intranasal esketamine [33] and subcutaneous esketamine [34] in patients with TRBD; the remaining of the 9 studies examined administered IV ketamine concomitant with an existing mood stabilizers/antidepressant (Table 1)”.
Discussion:
Anhedonic effect was also replicated (doi: 10.3389/fpsyt.2021.704330) by Wilkowska et al. (2021) and might ameliorate the observation mentioned in the discussion.
Response: Thank you for this comment. We have added to the discussion- page 12 line 392- the following sentence which includes the Wilkowska et al. (2021):” A recent study by Wilkowska and colleagues showed a significant reduction in anhedonia across ketamine infusions (p<0.001) in patients with unipolar and bipolar treatment resistant depression, however, they did not provide separate data for patients with TRBD [38].”.
Conclusions:
I strongly encourage elaborating on the implication for the future research with regard to dose, formulation, administration frequency and treatment duration.
Response: Thank you for this suggestion. We have included the following text in the conclusion-page 13, line 477- which now reads:” Moreover, future studies comparing different ketamine/esketamine routes of administration, as well as different dosages and length of treatment are needed to solidify the evidence”.
Reviewer 2 Report
Comments and Suggestions for Authors
In my opinion, the manuscript entitled “An update on the efficacy of single and serial intravenous ketamine infusions and esketamine for bipolar depression: a systematic review and meta-analysis” is generally very well organized and well-written. The chosen subject is interesting and important. However, several issues have to be explained/added:
* In the Introduction section the Authors should briefly present available treatment strategies for bipolar depression.
* On the basis of the analyzed data, what kind of recommendation would the Authors give to clinicians in relation to ketamine and esketamine use in patients with bipolar depression?
* The manuscript should be checked one more time in order to correct typos.
Comments on the Quality of English LanguageEnglish language is fine.
Author Response
Dear reviewer,
We appreciate your efforts and the thoughtful critiques. We have addressed every comment to the best of our ability. Our point-by-point responses are listed below.
Reviewer 2:
Comments to the Author:
In my opinion, the manuscript entitled “An update on the efficacy of single and serial intravenous ketamine infusions and esketamine for bipolar depression: a systematic review and meta-analysis” is generally very well organized and well-written. The chosen subject is interesting and important. However, several issues have to be explained/added:
* In the Introduction section the Authors should briefly present available treatment strategies for bipolar depression.
Response: Thank you for this comment. We have now expanded the introduction-page 2 starting at line 53- adding sentences regarding treatment strategies for bipolar depression which now read: “Current pharmacological options for bipolar depression include cariprazine, lumateperone, lurasidone, quetiapine, and olanzapine/fluoxetine combination. Lamotrigine, as a maintenance therapy, has shown the reduce the risk of future depressive episodes, although it takes almost 6-8 weeks to reach therapeu-tic dosage. Augmentation strategies with mood stabilizers, thyroid hormones, dopamine agonists such as pramipexole and psychostimulants have attempted to surpass the significant challenges of the depressive phase of the illness.”
* On the basis of the analyzed data, what kind of recommendation would the Authors give to clinicians in relation to ketamine and esketamine use in patients with bipolar depression?
Response: Thank you for this helpful comment. We have added a section on clinical consideration –page 13 line 462:
4.2 Clinical considerations
Patients with TRBD should carefully consider ketamine and esketamine for short duration while concurrently continuing a robust mood stabilizer in an acute depressive episode. The study by Fancy et al showed a more robust response among patients with BD-II compared to BD-I, thus, suggesting a possible option for BD-II. It is important to note that the prolonged use of ketamine carries a potential risk for treatment-emergent affective switching (TEAS) and risk of dependency, necessitating careful and thorough monitoring. Dose escalation studies beyond 0.5 mg/kg body weight for racemic keta-mine have not established in TRBD, thus, extreme caution is advisable if dose escalation is considered to minimize the risk of negative consequences. In the absence of an RCT investigating intranasal esketamine for TRBD, there is limited data to provide recommendation for esketamine use for TRBD, although the data from one observational study seems reassuring.
* The manuscript should be checked one more time in order to correct typos.
Response: Thank you for this comment. We have reviewed the text and corrected any misspelling or typos.